# Genotypic Homogeneity in Distinctive Transforming Growth Factor-Beta Induced (TGFBI) Protein Phenotypes

**DOI:** 10.3390/ijms22031230

**Published:** 2021-01-27

**Authors:** Sang Beom Han, Venkatraman Anandalakshmi, Chee Wai Wong, Si Rui Ng, Jodhbir S. Mehta

**Affiliations:** 1Department of Ophthalmology, Kangwon National University School of Medicine, Chuncheon 24289, Korea; msbhan@nate.com; 2Department of Ophthalmology, Kangwon National University Hospital, Chuncheon 24289, Korea; 3Singapore Eye Research Institute, Singapore 169856, Singapore; vidhyavraman@yahoo.com (V.A.); wongcheewai81@gmail.com (C.W.W.); sirui.ng@mohh.com.sg (S.R.N.); 4Singapore National Eye Centre, Singapore 168751, Singapore; 5Ophthalmology and Visual Sciences Academic Clinical Program, Duke-NUS Graduate Medical School, Singapore 169857, Singapore

**Keywords:** corneal dystrophy, granular corneal dystrophy, lattice corneal dystrophy, *TGFBI*, transforming growth factor beta induced protein, aggregation disorders

## Abstract

**Background:** To evaluate the distribution of the transforming growth factor-beta induced (TGFBI) corneal dystrophies in a multi-ethnic population in Singapore, and to present the different phenotypes with the same genotype. **Methods:** This study included 32 patients. Slit lamp biomicroscopy was performed for each patient to determine the disease phenotype. Genomic DNA was extracted from the blood samples and the 17 exons of the TGFBI gene were amplified by PCR and sequenced bi-directionally for genotype analysis. **Results:** Regarding phenotypes, the study patients comprised 11 (34.4%; 8 with R555W and 3 with R124H mutation) patients with granular corneal dystrophy type 1 (GCD1), 6 (18.8%; 5 with R124H and 1 with R124C mutation) patients with GCD2, 13 (40.6%; 7 with R124C, 2 with H626R, 2 with L550P, 1 with A620D and 1 with H572R) patients with lattice corneal dystrophy (LCD) and 2 (6.3%; 1 with R124L and 1 with R124C) patients with Reis–Bückler corneal dystrophy. Regarding genotype, R124H mutation was associated with GCD2 (5 cases; 62.5%) and GCD1 (3 cases; 37.5%). R124C mutation was associated with LCD (7 cases; 87.5%) and GCD2 (1 case; 12.5%). All the 8 cases (100%) of R555W mutation were associated with GCD1. **Conclusions:** Although the association between genotype and phenotype was good in most cases (65.7%; 21 of 32 patients), genotype/phenotype discrepancy was observed in a significant number.

## 1. Introduction

Corneal dystrophies represent a group of hereditary, bilateral and non-inflammatory conditions characterized by the progressive accumulation of abnormal deposits in different layers of the cornea, which causes corneal opacity and visual impairment [1,2]. Mutations in the transforming growth factor β-induced (*TGFBI*) gene on chromosome 5q31 result in the https://paperpile.com/c/Lr4V24/3N6Nproduction and extracellular accumulation of mutated abnormal TGFBI protein [3,4], which leads to superficial and stromal corneal dystrophies [2].

Munier et al. [4] first recognized the associations between the phenotypes and genotypes for corneal dystrophies caused by *TGFBI* gene mutations, as follows: p.Arg124Leu(R124L) for Reis–Bücklers corneal dystrophy (RBCD), p.Arg555Gln (R555Q) for Thiel-Behnke corneal dystrophy (TBCD), p.Arg555Trp (R555W) for granular corneal dystrophy type 1 (GCD1), p.Arg124His (R124H) for granular corneal dystrophy type 2 (GCD2) and p.Arg124Cys (R124C) for lattice corneal dystrophy type 1 (LCD1) [4]. However, classification of various types of *TGFBI* corneal dystrophies based on clinical findings has remained challenging because there is not a one-to-one correspondence between phenotypic appearances and genetic mutations [5].

In 2008, the International Committee for Classification of Corneal Dystrophies (IC3D) introduced a new classification system for corneal dystrophies by integrating information on genotypes as well as phenotype and pathology [6]. The IC3D further updated the classification in 2015 that incorporated new clinical, histopathologic and genetic information [7], dividing the dystrophies into epithelial and subepithelial dystrophies, epithelial-stromal *TGFBI* dystrophies including RBCD, TBCD, GCD1, GCD2 and LCD, stromal dystrophies and endothelial dystrophies, according to the corneal layer primarily involved [7].

Previous studies have suggested that a strong correlation between phenotypic change and genotypic mutations for the majority of *TGFBI* corneal dystrophies including GCD and LCD [8,9,10]. However, there are mutation hot spots e.g., at codon R124 and R555 that are responsible for majority of cases [8]. Due to this, there are more reports of discrepancy between genotype and phenotype [11,12,13,14]. Hence, the same genotype can cause a large variation in phenotypes [10,14,15], likewise the same phenotypical presentation has been shown to be caused by multiple different genotypic mutations [12,13,16,17].

In this study, we aimed to evaluate the distribution of the phenotypes and genotypes of the *TGFBI* mutations in Singapore, a nation with a multi-ethnic population and to determine the genotype/phenotype correlation of the diseases.

## 2. Patients and Methods

### 2.1. Patients and Blood Sample Collection

This study was approved by the Singhealth Centralised Institutional Review Board and adhered to the tenets of the Declaration of Helsinki. A consecutive series of unrelated 32 patients who were clinically diagnosed as having *TGFBI* corneal dystrophies at Singapore National Eye Centre from 1999 to 2019 were retrospectively enrolled. All the 32 patients underwent *TGFBI* gene analysis.

Slit lamp biomicroscopy and anterior segment photography was performed to evaluate and document the depth, size, shape and distribution of the corneal deposits. Anterior segment optical coherence tomography (AS-OCT; Carl Zeiss Meditec, Jena, Germany) was also done to assess depth of the lesions. Approximately 5 mL of blood were collected from patients in EDTA tubes. Blood samples were kept frozen at −20 °C until DNA was extracted. Written informed consent were obtained from all patients prior to blood collection. Ethical approval for the collection of blood samples was granted by the Singhealth Institutional Review Board.

### 2.2. PCR Amplification Reaction and Sequencing

Genomic DNA was extracted from peripheral blood leukocytes of each patient using the Nucleon^®^ blood extraction kit (Tepnel, Manchester, UK). The 17 exons of *TGFBI* gene were amplified by Polymerase Chain Reaction (PCR) using the primers as described previously (Table 1) [18]. The PCR products were purified using PCR clean-up kits (Axygen Biosciences, Union City, CA, USA) and sequenced using Big Dye Terminator v3.1 chemistry (Applied Biosystems, Foster City, CA, USA). Bi-directional sequencing of the amplicons was carried out on an ABI prism 3100 genetic analyzer (Applied Biosystems). The obtained nucleotide sequences were analyzed for base-pair changes using the Lasergene V.8.0 software (DNASTAR, Inc., Madison, WI, USA).

The original published *TGFBI* cDNa sequence (GenBank accession no. NM_000358) was used for sequence comparison. Numbering of the base pair change for identification of mutations was based on the original reference sequence with +1 corresponding to the A of the translation initiation codon ATG. For 8 of the reported patients, whole exome sequencing (WES) was performed at 3 billion, Inc (Seoul, South Korea), using genomic DNA isolated from the patient’s whole blood with informed consent obtained from the patient.

## 3. Results

### Distribution of Genotypes and Phenotypes

Thirty-two patients with an average age of 61.4 ± 18.1 years (mean ± SD; range, 22–94) were included in the study. Regarding the genetic mutations, a total of nine mutations in the *TGFBI* gene, i.e., *R124H, R124C* and *R124L* in the first fasciclin 1 (FAS1) domain and, *R555W*, *R555Q*, *H262R*, *H572R*, *A620D* and *L550P* in the fourth FAS1 domain were detected.

Regarding the phenotypes of *TGFBI* corneal dystrophy, the subjects comprised 11 (34.4%) patients with GCD1, 6 (18.8%) patients with GCD2, 13 (40.6%) patients with LCD, 2 (6.3%) patients with RBCD. Regarding the distribution of *TGFBI* mutations in each phenotype of corneal dystrophy, patients with GCD1 consisted of 3 patients with R124H and 8 patients with R555W, and those with GCD2 comprised 5 patients with R124H and 1 patient with R124C. Patients with LCD comprised 7 patients with R124C, 2 patients with H626R, 2 patients with L550P, 1 patient with A620D and 1 patient with H572R. For 2 patients with RBCD, 1 patient had R124L mutation and 1 patient had R555Q mutation (Table 2). Table 3 shows the distribution of corneal dystrophies among the different ethnicities, which suggests possible distinct ethnic clustering of several mutations, although sample size was small. GCD1 with R555W mutation was the most common dystrophy found in patients of Indian ethnicity, whereas LCD with R124C mutation was the prominent dystrophy amongst patients of Malay ethnicity. The Chinese population showed a heterogeneous mix of *TGFBI* corneal dystrophies, in which GCD1 was the most common, followed by GCD2, LCD and RBCD (Table 2).

Regarding the distribution of *TGFBI* mutations, the mutation of the codon of R124 (17 cases, 53.1%) was the most common, followed by the mutation of the locus R555 (9 cases, 28.1%). The R124H mutation (8 cases) was associated with GCD1 in 3 cases (37.5%) and GCD2 in 5 cases (62.5%). In cases associated with R124H mutation, clinical findings varied from typical findings of GCD1, such as bread crumb like lesions with clear intervening stroma that do not extend to the periphery, to lesions typically seen in GCD2, i.e., discoid lesions and star shaped, spiky deposits (Figure 1), The R124C mutation (8 cases) caused LCD in 7 cases (87.5%) and GCD2 in 1 case (12.5%). In LCD cases associated with R124C mutation, the phenotypic alterations ranged from subepithelial haze only, to central subepithelial haze together with feathery opacities, thin lattice lines, white dots and flecks, or a combination of these. In 1 case of the R124H mutation, lattice lines with multiple grayish white dots and some spiky deposits were observed (Figure 2). The R555W mutation led to GCD1 in all 8 cases (100%). However, large variation of phenotypic changes were found, i.e., discoid lesions with central clearing, white dots and bands, star shaped whitish lesions and dots with hazy intervening stroma, translucent specks or a mixture of grayish translucent specks and denser whitish dots and bands (Figure 3). One case with the R555Q mutation was associated with RBCD. The other mutations, including H626R, A620D, L550P and H572R were all associated with classic manifestations of LCD, such as, similar refractive lines, dots and subepithelial haze of varying severity LCD (Table 4, Figure 4) [2].

## 4. Discussion

The present study demonstrated that the mutations in the codons of R124 and R555, such as, R124H, R555W and R124C were the most common mutations in a multi-ethnic population in Singapore, corroborating well with the results of previous studies in different countries [2,5,8,19,20,21,22,23,24]. The predominance of R124H, R555W and R124C mutations in various geographical locations worldwide indicates that these mutations may represent mutation hot spots in the gene [2]. The differences in phenotypes and genotypes among different ethnicities in Singapore in the present study suggest that the frequency of *TGFBI* corneal dystrophies can vary among different races, although this study included relatively small number of patients. Other studies have also suggested the possible differences in frequency of *TGFBI* corneal dystrophies in different areas around the world [8,9,10,21,24,25,26,27,28,29,30].

Studies in East Asian countries including Korea and Japan showed striking predominance of R124H mutation and GCD2 [21,25,26,27,28], whereas studies in Chinese population suggested the prevalence of R124C, R555W and R124H mutations were comparable [10,31]. Studies in India demonstrated higher proportion of R124C and R555W mutations, whereas R124H mutation was relatively rare [29,30]. The distribution of expressivity of *TGFBI* corneal dystrophies in Singaporeans, a multi-ethnic population comprising Chinese, Indian, Malaysian and others, might reflect these differences. Further studies with larger patient group are needed for evaluation of the influence of ethnicities on expressivity of *TGFBI* corneal dystrophies.

In general, the results of this study indicated moderate genotype/phenotype correlation in *TGFBI* corneal dystrophies, such as, R555W and GCD1, R124H and GCD2, and R124C and LCD, which corroborate with previous studies [2,5,8,9,10,23,32]. R555W and R124H mutations shows strong association with GCD1 and GCD2, respectively, and 99% of the genotypic mutations led to corresponding phenotypic changes [2,9,10,27,28]. R124C mutation also has close correlation with LCD [2,23,32]. However, the results also revealed that the discrepancy between phenotypes and genotypes can also exist, i.e., the same mutation can result in different phenotypes and similar phenotypes can be caused by various mutations [5]. In this study, R124H mutation was associated with GCD1 in 2 cases, R124C was associated with GCD2 in 1 case, and R124L was associated with RBCD instead of TBCD in 1 case. In addition, there was a large variation in clinical changes even in the same phenotypes associated with the same genotypic mutations (Figure 1, Figure 2 and Figure 3). LCD phenotype was associated with various mutations, i.e., H626R, A620D, L550P, H572R as well as R124C (Figure 4), suggesting that these mutations can lead to increased propensity for the formation and accumulation of amyloid fibrils [5].

Previous studies have shown that R124C mutation can also be associated with GCD2, RBCD and TBCD, although the mutation is most commonly associated with LCD [12,13,16,17,33]. Cases with subepithelial granular aggregates as well as stromal lattice deposits in association with R124C gene mutation have been reported [13,33], and this phenomenon was also observed in one patient of the present study. Moreover, R124C mutation was also reported to be associated to RBCD [16,17] and TBCD [12]. Conversely, LCD can be caused by many different mutations of *TGFBI* gene, as shown in this study [2,5]. A620D mutation was also found to be associated with LCD in a study in Korea [34]. Studies in other countries have also shown H626R mutation can lead to LCD with marked phenotypic heterogeneity [9,35]. A further mutation at the same base pair resulting in H626P has also been described in LCD as well as in significantly different phenotypes that were clinically similar to TBCD and RBCD [36,37], suggesting that mutations at this site can result in various phenotypes [5]. In addition to the mutations observed in this study, a large variety of mutations, e.g., A546T, L569Q, T621P, L527R and P542R, which are in the fourth FAS1 domain were demonstrated to be associated with LCD [27,28,38], suggesting that these mutations can also have high capacity for the amyloidogenesis [5].

In the present study, R124L and R555Q mutations were found in RBCD cases, although the mutations are typically associated with RBCD and TBCD, respectively [2]. A study in Brazil also reported that RBCD could be caused by R555Q and R124L [38]. Zeng et al. [39] showed that R124L and R555Q can also be associated with GCD1 and GCD2, which also support possible discrepancy between genotype and phenotype in *TGFBI* corneal dystrophies [5].

Regarding the mechanism underlying this possible discrepancy, a role of interplay of genetic and environmental factors has been suggested [10,14], i.e., modifications from other genes and environmental circumstances might have influence on penetrance and expressivity of the mutated genes [10,11,14,15]. Protein production at a local level in the cornea might be modulated by other genetic factors that might affect the abundance of TGFBI protein or the function of the protein processing pathway [10,11,14,15]. For instance, contact lens wear was suggested to be a possible anti-aggravation factor and to prevent progression and recurrence of corneal deposits in patients with GCD2 [14,40]. Although *TGFBI* corneal dystrophies are autosomal dominant hereditary disorders, R124H heterozytes have been shown to tend to show less severe corneal opacity than homozygotes [2,11,41,42]. R124H heterozygotes can sometimes have incomplete penetrance, which may lead to resemblance to the phenotype of GCD1, e.g., granular deposits without lattice opacities [10,11,43,44]. Song et al. [44] reported a case with the heterozygous R124H mutation in which no corneal opacities were observed, suggesting the mutation can even have non-penetrance, which can have important implication for refractive surgery in phenotypically normal patients with *TGFBI* mutation [44]. Three patients with phenotype of GCD1 in association with R124H mutations were also observed in the present study, conceivably due to the reduced penetrance. Among the 8 patients with R124H mutation in the present study, zygosity data were available in 4 patients, all the 4 patients were heterozygote, in which 3 had GCD2 and 1 had GCD1.

Disturbance of genotype/phenotype correlation in some cases may suggest that the scheme for nomenclature and classification of the *TGFBI* corneal dystrophies is still imperfect [5], indicating that development of a modified classification system [2]. It can also be considered to check the presence of typical correspondence between genotype and phenotype, and if the discrepancy is found, then evaluate the possible secondary factors, i.e., homo/heterozygosity, contact lens wear, history of corneal erosion, ethnic variation and herpes simplex keratitis (Figure 5).

The limitations of this study are as follows: (1) This study did not include the pathologic examination of the cornea specimen. Further studies that include histopathologic analysis of the corneal samples are needed for the investigation of the pathophysiology underlying the association between genotype and phenotype of the corneal dystrophies. (2) This study included relatively small sample size, which may not be sufficient for the evaluation of the genotype/phenotype correlation and influence of the ethnicity. Thus, we believe further studies with larger population are needed to develop a repository of cases for evaluation of these in detail. (3) We cannot completely rule out the possibility that incorrect phenotypic and genotypic description could cause misclassification of the subtype. Scientific phenotypical/genotypical relationship of the *TGFBI* corneal dystrophies may only be possible in forthcoming prospective studies in which corneal presentation is thoroughly recorded in direct and indirect illumination and by pharmacologically dilated pupil and focusing the epithelium, the stroma and the endothelium, and the minimum of five patients in three generations in one family are explored.

In conclusion, we have shown the distribution of genotype and phenotype of *TGFBI* corneal dystrophies in a multi-ethnic population in Singapore. Although genotype/phenotype correlation was moderate, discrepancy was observed in cases, which warrant further studies for elucidation of the pathophysiology underlying these association.

## Figures and Tables

**Figure 1 ijms-22-01230-f001:**
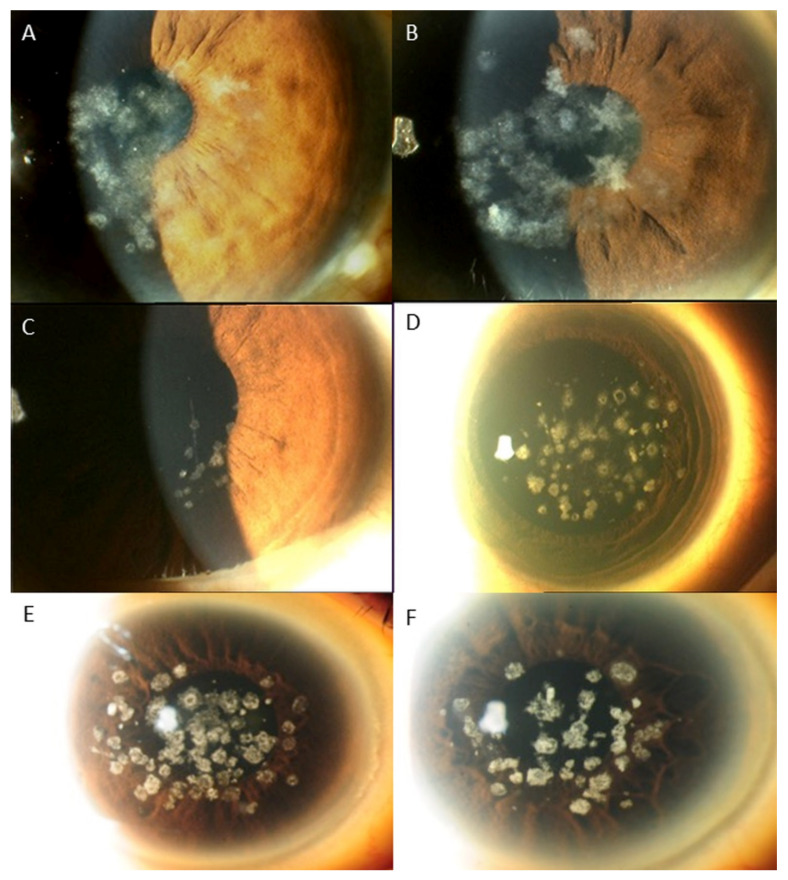
The phenotypic variations associated with mutation at R124H: (**A**): Classic bread crumb like whitish lesions in the anterior stroma that can be seen GCD1; (**B**): Similar to A but a few lesions have stellate pattern that is seen in GCD2; (**C**): Few translucent dots that can be seen in GCD1; (**D**): multiple grayish white dots, some with moth eaten central clearing and some spiky deposits that can be seen in GCD 2; (**E**): classic bread crumb like lesions with clear intervening stroma that can be seen in GCD1; (**F**): similar bread crumb like lesions interspersed with some spiky deposits that can be seen in GCD2.

**Figure 2 ijms-22-01230-f002:**
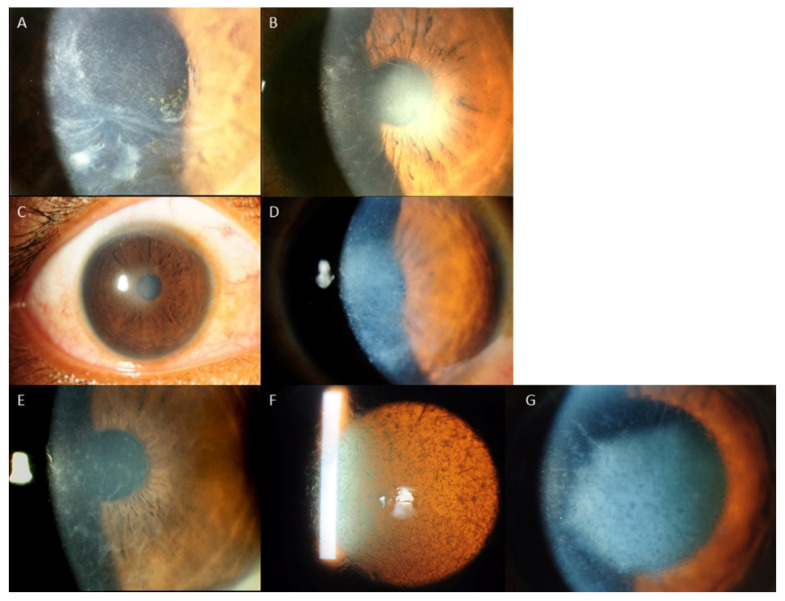
Different phenotypes of LCD1 associated with the same *R124C* mutation: (**A**): subepithelial haze and feathery opacities; (**B**): Thin lattice lines with subepithelial haze; (**C**): Central subepithelial haze only; (**D**): Central subepithelial haze with white dots and few lattice lines; (**E**): Dots and fleck like deposits; (**F**,**G**): Lattice lines with multiple grayish white dots and some spiky deposits (GCD2).

**Figure 3 ijms-22-01230-f003:**
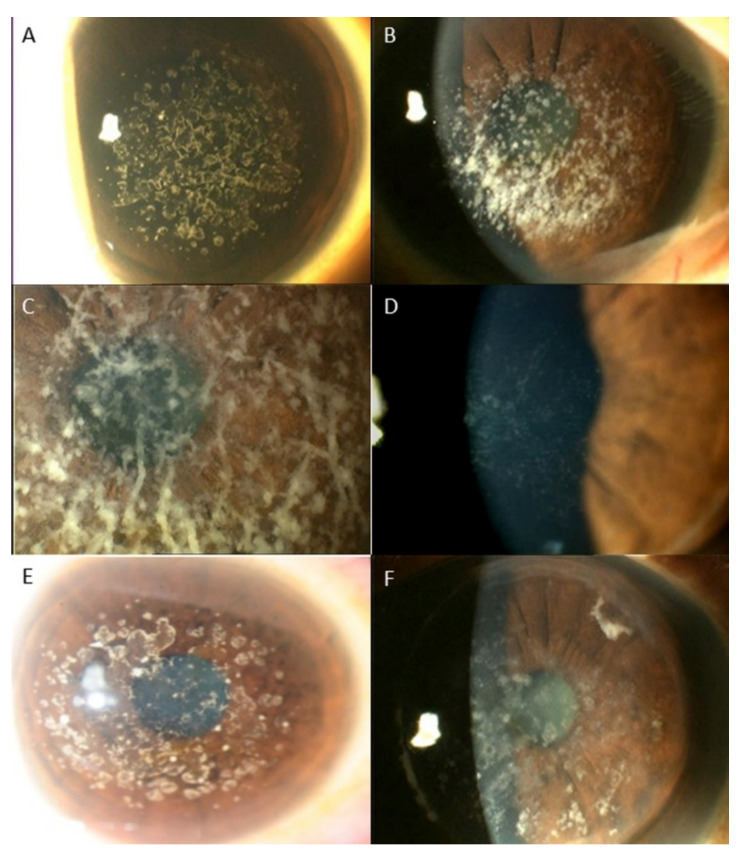
Different phenotypes of GCD1 associated with the same *R555W* mutation: (**A**): discoid lesions with central clearing; (**B**): White dots of varying densities, some coalescing to form white bands; (**C**): Star-shaped whitish lesions and dots with hazy intervening stroma; (**D**): Specks of translucent dots; (**E**): bread crumb like lesions sparing the peripheral cornea; (**F**): Grayish specks centrally and denser white dots that coalesce in the peripheral cornea.

**Figure 4 ijms-22-01230-f004:**
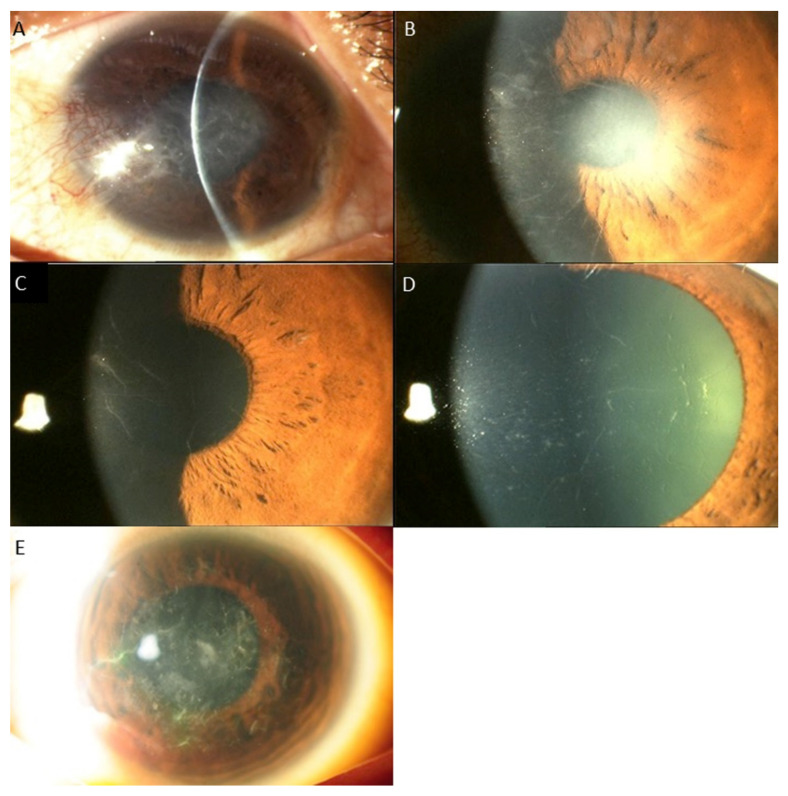
LCD1 phenotypes associated with 5 different mutations demonstrating similar phenotypic change, i.e., similar refractive lines, dots and subepithelial haze of varying severity. (**A**): *H572R*, (**B**): *R124C*, (**C**): *H626R*, (**D**): *L550P*, (**E**): *A620D*.

**Figure 5 ijms-22-01230-f005:**
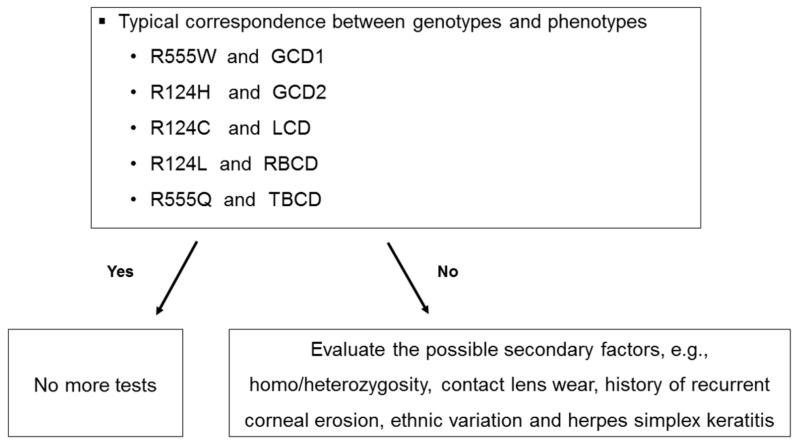
Flowchart for the genotype/phenotype correlation of *TGFBI* corneal dystrophies. The presence of typical correspondence between genotype and phenotypes can be checked, and if the discrepancy is found, the possible secondary factors, such as, homo/heterozygosity, contact lens wear, history of recurrent corneal erosion, ethnic variation and herpes simplex keratitis, can be evaluated.

**Table 1 ijms-22-01230-t001:** Primers for the TGFBI DNA sequencing.

Exon	Forward Primer	Reverse Primer	Product Size (bp)
1	CGGAGGCGCTCTCACTTCC	CGAGCCCCGACTACCTGACC	265
2	GGGAGTCATTAAAGTGGGGTGGA	AGCTTGGTCTCCTGGCTGGTTAC	99
3	CAACTTAGTGGAGAGGGGCCAGA	CTCTCTCCCACCATTCCCTTCC	206
4	GCCATCCCTCCTTCTGTCTTCTG	CCGGGCAGACGGAGGTCATC	217
5	ACTGACACCCTGTCCTTCCTCCT	AGCCCACACATGGAACAGAAATG	261
6	CTGCTCATCCTTGCTGCTTCTCT	AGAGTTCCTGCTAGGCCCCTCTT	249
7	TCTGTGGGGAGTGCCAGAGTC	CAAATGAGGCAGCAGCAGGA	234
8	TGGACCCTGACTTGACCTGAGTC	AAAGGATGGCAGAAGAGATGGTG	311
9	CCCTGGGGTGGATGAATGATAAA	GCCTCCAGGGACAATCTAACAGG	251
10	ATTGCAGGAGCACATCTCTCTGG	GCTTCCCAGGAGCATGATTTAGG	230
11	GCCCCTCGTGGAAGTATAACCAG	ATCCCACTCCAGCATGACCACT	248
12	TGACAGGTGACATTTTCTGTGTGTG	GGGCCCTGAGGGATCACTACTTT	224
13	TGACCAGGCTAATTACCATTCTTGG	CAGCCTTTGATTTGCAGGACACT	210
14	TGCTCTACTTTCAACCACTACTCTG	CCAACTGCCACATGAAGAAAAGG	198
15	TCACTCTGGTCAAACCTGCCTTT	CCTCTATGGCCCAAACAGAGGAC	183
16	GCCATTGTCATAAGCAGTTGCAG	ATACAGCAGATGGCAGGCTTGG	176
17	TGGGGAGATCTGCACCTATTTGA	GGTCAGCACACTGTACCATGCAC	710

**Table 2 ijms-22-01230-t002:** The distribution of genotypes of *TGFBI* mutations in each phenotype of TGFBI corneal dystrophy.

Phenotype	Associated *TGFBI* Mutations (Genotypes)
Granular corneal dystrophy (GCD), type 1 (n = 11)	R124H (n = 3)
	R555W (n = 8)
Granular corneal dystrophy (GCD), type 2 (n = 6)	R124H (n = 5)
	R124C (n = 1)
Lattice corneal dystrophy (LCD) (n = 13)	R124C (n = 7)
	H626R (n = 2)
	L550P (n = 2)
	A620D (n = 1)
	H572R (n = 1)
Reis-Buckler dystrophy (RBCD) (n = 2)	R124L (n = 1)
	R555Q (n = 1)

**Table 3 ijms-22-01230-t003:** Phenotypes and genotypes of *TGFBI* corneal dystrophy in various ethnicities.

Ethnicity	Phenotype	Associated *TGFBI* Mutations
Chinese (n = 16)	GCD1 (n = 5)	R124H (n = 3)
		R555W (n = 2)
	GCD2 (n = 5)	R124H (n = 4)
		R124C (n = 1)
	LCD (n = 4)	R124C (n = 1)
		A620D (n = 1)
		H626R (n = 2)
	RBCD (n = 2)	R124L (n = 1)
		R555Q (n = 1)
Malay (n = 6)	LCD (n = 6)	R124C (n = 6)
Indian (n = 5)	GCD1 (n = 5)	R555W (n = 5)
Others (n = 5)	GCD1 (n = 1)	R555W (n = 1)
	GCD2 (n =1)	R124H (n = 1)
	LCD (n = 3)	L550P (n = 2)
		H572R (n = 1)

**Table 4 ijms-22-01230-t004:** The distribution of phenotypes of TGFBI corneal dystrophies in each genotype of TGFBI mutations.

Locus	*TGFBI* Mutations	Phenotype
R124 (n = 17)	R124H (n = 8)	GCD1 (n = 3)
		GCD 2 (n = 5)
	R124C (n = 8)	GCD2 (n = 1)
		LCD (n = 7)
	R124L (n = 1)	RBCD (n = 1)
R555 (n = 9)	R555W (n = 8)	GCD1 (n = 8)
	R555Q (n = 1)	RBCD (n = 1)
Others (n = 6)	H626R (n = 2)	LCD
	A620D (n = 1)	LCD
	L550P (n = 2)	LCD
	H572R (n = 1)	LCD

## Data Availability

The data presented in this study are available on request from the corresponding author. The data are not publicly available due to privacy of the study subjects.

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
