# Peer review of "Genotypic Homogeneity in Distinctive Transforming Growth Factor-Beta Induced (TGFBI) Protein Phenotypes"

_ijms, 2021, doi:10.3390/ijms22031230_

Round 1
Reviewer 1 Report
The authors report in their retrospective IJMS-1088190 manuscript about 32 patients with TGFBI corneal dystrophy and 4 patients without DNA analysis. It is a multi-ethnic study of Singapore`s population. Genotype/phenotype discrepancy was observed in a significant number.
My specific comment are as follows:
1. It is an interesting retrospective study.
2. However, we need correct rules to compare the phenotypical/genotypical
relationship of the epithelial-stromal TGFBI corneal dystrophies.
3. The rules generally apply to all forms of corneal dystrophy:
3.1. Corneal presentation of an affected patient at the slit lamp in direct and
indirect illumination and by pharmacologically dilated pupil and focussing
the epithelium, the stroma and the endothelium. DNA analysis.
3.2. We need a minimum of five patients and a minimum of three
generations in one family to explore 3.1. and the DNA analysis.
4. To the authors: the four patients of your study with unknown mutation are
to remove.
5. The incorrect phenotypical and genotypical description above all of the
TGFBI corneal dystrophies caused a lot of misinterpretation in the past
literature. We know the inter-and intrafamiliar phenotypical differences of
the TGFBI corneal dystrophies due to the distinct penetrance of the gene.
I give you 2 examples: You cited in your manuscript the two publications of
Chang et al and Yang et al. who described patients with RBCD and TBCD
and the LCD mutation of R124C. However, these authors presented to their
patients only corneal photos in direct illumination and by use pupil.
What about photos in indirect illumination and by dilated pupil and focussing
the stroma: lattice lines !?
6. I suggest that the authors declare that a scientific phenotypical/
genotypical relationship of the TGBBI corneal dystrophies is only possible in
forthcoming prospective studies in form of the rules in point 3.1 and 3.2.
7. In your study the following patients are of special interest:
the 3 GCD1 patients with R124H,
1 GCD2 patient with R124C,
6 LCD patients with new mutations (H626R and ---)
As you know we distinguish between LCD1 and variants (LCD3 and
LCD3a). LCD2 is a misnomer because it is a systemic inherited amyloidosis.
8. The epidemiologic distribution of your patients is of interest. We know that
GCD2 is more frequent than GCD1 in Japan and South Korea. In China
GCD1 is more frequent than GCD2.
9. I suggest a minor revision of your manuscript.
Author Response
1. It is an interesting retrospective study.
⇒ Thank you very much for the encouraging comments.
2. However, we need correct rules to compare the phenotypical/genotypical relationship of the epithelial-stromal TGFBI corneal dystrophies.
⇒ Yes, we agree with you that correct rules, i.e. rules in point 3.1 and 3.2 are need to compare the phenotypical/genotypical relationship of the epithelial-stromal TGFBI corneal dystrophies. We mentioned that forthcoming prospective studies with correct rules are therefore needed (Limitation (3), Page 12).
3. The rules generally apply to all forms of corneal dystrophy:
3.1. Corneal presentation of an affected patient at the slit lamp in direct and indirect illumination and by pharmacologically dilated pupil and focusing the epithelium, the stroma and the endothelium. DNA analysis.
3.2. We need a minimum of five patients and a minimum of three generations in one family to explore 3.1. and the DNA analysis.
⇒ We agree with your points, and have added that phenotypical / genotypical relationship of TGFBI corneal dystrophies is only possible in forthcoming prospective studies in form of the rules in point 3.1 and 3.2 in the Discussion (Limitation (3), Page 12).
4. To the authors: the four patients of your study with unknown mutation are to remove.
⇒ As you recommended, we removed the four patients with unknown mutation, and revised the manuscript accordingly.
5. The incorrect phenotypical and genotypical description above all of the TGFBI corneal dystrophies caused a lot of misinterpretation in the past literature. We know the inter-and intrafamiliar phenotypical differences of the TGFBI corneal dystrophies due to the distinct penetrance of the gene.
I give you 2 examples: You cited in your manuscript the two publications of Chang et al and Yang et al. who described patients with RBCD and TBCD and the LCD mutation of R124C. However, these authors presented to their patients only corneal photos in direct illumination and by use pupil. What about photos in indirect illumination and by dilated pupil and focussing the stroma: lattice lines !?
⇒ We agree with you that the incorrect phenotypic and genotypic description might cause misinterpretation, and mentioned it in the revised manuscript (Limitation (3), Page 12).
6. I suggest that the authors declare that a scientific phenotypical/ genotypical relationship of the TGBBI corneal dystrophies is only possible in forthcoming prospective studies in form of the rules in point 3.1 and 3.2.
⇒ As the reviewer suggested, we have added that a scientific phenotypic / genotypic relationship of TGFBI corneal dystrophies is only possible in forthcoming prospective studies, in the form of the rules in point 3.1 and 3.2 (Limitation (3), Page 12).
7. In your study the following patients are of special interest: the 3 GCD1 patients with R124H, 1 GCD2 patient with R124C, 6 LCD patients with new mutations (H626R and ---) As you know we distinguish between LCD1 and variants (LCD3 and LCD3a). LCD2 is a misnomer because it is a systemic inherited amyloidosis.
⇒ Thank you very much. We also think the patients are the most important points of the present study.
8. The epidemiologic distribution of your patients is of interest. We know that GCD2 is more frequent than GCD1 in Japan and South Korea. In China GCD1 is more frequent than GCD2.
⇒ Thank you very much. We also believe the epidemiologic distribution is of interest and warrant further studies.
9. I suggest a minor revision of your manuscript.
⇒ Thank you very much. We have performed a minor revision of the manuscript, according to the reviewer’s recommendations.

Reviewer 2 Report
Overall, a well-done study examining genotype-phenotype correlations in stromal corneal dystrophies. The methods are sound and well-described. The images are excellent and assist the reader in appreciating the phenotype for themselves.
In papers such as this, being open with data can be quite helpful and the authors have done a nice job sharing both the images, as described above, and the genetic sequences. In the future, other studies will be able to draw on these data.
One change - for Figure 2F, the retroillumination image is excellent and makes the opacities clear. However, given that the other images are all direct illumination, it would be helpful to see what direct illumination looks like for this eye as well.
Author Response
One change - for Figure 2F, the retroillumination image is excellent and makes the opacities clear. However, given that the other images are all direct illumination, it would be helpful to see what direct illumination looks like for this eye as well.
⇒ As the reviewer requested, we added Fig 2G, a direct illumination image of the patient with retroillumination image of Fig 2F.
